POINT OF VIEW

# Four erroneous beliefs thwarting more trustworthy research

**Abstract** A range of problems currently undermines public trust in biomedical research. We discuss four erroneous beliefs that may prevent the biomedical research community from recognizing the need to focus on deserving this trust, and thus which act as powerful barriers to necessary improvements in the research process.

**MARK YARBOROUGH\*, ROBERT NADON AND DAVID G KARLIN**

**Competing interests:** The authors declare that no competing interests exist.

## Introduction

In 2014, in an essay titled 'Why scientists should be held to a higher standard of honesty than the average person,' a former editor of the British Medical Journal argued that science depends wholly on trust (*Smith, 2014*). While many in the biomedical research community may quibble over the word 'wholly' here, few would dispute his overall point: the public's confidence is essential to the future of research. According to a noted scholar on the subject, the best way to enjoy trust is to deserve it (*Hardin, 2002*). One would hope that the research community is a deserving case, given the existence of safeguards such as professional norms, regulatory compliance and peer review. Unfortunately, there is an ever-growing body of evidence that calls into question the effectiveness of these measures.

This evidence includes, but is by no means limited to, findings about underpowered studies (*Ioannidis, 2005*), routine overestimations of efficacy (*Sena et al., 2010*; *Tsilidis et al., 2013*), the failure to take prior research into account (*Robinson and Goodman, 2011*; *Lund et al., 2016*), a propensity to confuse hypothesis-generating studies with hypothesis-confirming ones (*Kimmelman et al., 2014*), a worrisome waste of resources (*Chalmers and Glasziou, 2009*), and the low uptake of critical reforms meant to improve research (*Enserink, 2017*; *Peers et al.,*

*2014*). A recent popular book, *Rigor Mortis*, synthesizes such evidence into a compelling narrative that casts the reputation of research in a negative light (*Harris, 2017*).

While all of this evidence is cause for concern, we are most concerned by the reluctance of the research community to implement the reforms that could improve research quality. One can imagine a continuum of research practices that impact how scientific understanding advances. At one end one encounters the unforgivable, such as data fabrication or falsification. At the other end one finds the perfect, such as published research reports so thorough that findings can be easily reproduced from them.

The concerns of interest to us in what follows have little to do with the misconduct found on the unforgivable end of the continuum. Instead, they fall all along it and pertain to unsound research practices (such as non-robust reporting of methods, flawed study designs, incomplete reporting of data handling, and deficient statistical analyses) that nevertheless impede the advance of science. These are the practices that reform measures could counter if researchers were less reluctant to adopt them. In an effort to account for this reluctance, we review four erroneous beliefs that we think contribute to it.

We acknowledge that we lack extensive data confirming the prevalence and distribution of these beliefs. Thus, readers can form their own

opinions about whether the beliefs are as widespread as we fear they are. We have come upon our concerns as a result of our careers related to biomedical research, which will be the focus of our remarks below, though we think the issues are relevant to life sciences research more broadly. One of us (MY) has extensively studied how to promote trustworthiness in biomedical research, and another (RN) has a long and successful career devoted to understanding the role of sound methodologies in producing it. The final author (DGK) is a preclinical researcher who was among those who pioneered early efforts to learn how researchers and research institutions can meaningfully connect the research community with the publics it seeks to serve. We think this collective pedigree lends credence to our analysis and to the strategy for moving forward that we recommend in the conclusion.

## Recognizing the barriers to a greater focus on deserving trust

### It's about the science, not the scientists

Erroneous belief one is that questioning the trustworthiness of research simultaneously questions the integrity of researchers. As a result, many individuals react counterproductively to calls to improve trustworthiness. They are akin to pilots who confuse discussions about improving the flightworthiness of airplanes with criticism of their aviation skills. Though understandable, such concerns miss the point (*Yarborough, 2014a*). The multitude of methods, materials, highly sophisticated procedures and complex analyses intrinsic to biomedical research all create ways for it to err, making it exceptionally difficult to detect problems (*Hines et al., 2014*). These are the critical matters that all researchers must learn to direct their attention to. Yet they cannot do so if constructive criticism about how to improve science is taken personally.

### We need to focus on the health of the orchard, not just the bad apples in it

Erroneous belief two is that the bulk of problems in research is due to bad actors. There is no doubt that misconduct is a substantial problem (*Fang et al., 2012*). This should not blind us, however, to how common study design and data analysis errors are in biomedical research (*Altman, 1994*). Indeed, these errors are likely

to increase due to trends in current scientific practice, particularly the growing size and interdisciplinarity of investigative teams (*Wuchty et al., 2007*; *He and Zhang, 2009*; *Gazni et al., 2012*). Because they require divisions of labor and expertise, such collaborations create fertile ground for producing unreliable research. Affected publications draw much less scrutiny than those of authors who engage in misconduct (*Steen et al., 2013*), and thus problems in them are likely to be discovered much later, if at all. For example, consider that the number of retracted publications is much less than 1% of published articles (*Grieneisen and Zhang, 2012*), yet publication bias has been found to affect entire classes of research (*Tsilidis et al., 2013*; *Macleod et al., 2015*).

The prevalence of erroneous research results and the enduring problems they cause require proactive efforts to detect and prevent them. What we find instead is a disproportionate emphasis on detecting and punishing 'bad apples.' The more we concentrate on this, the more difficult it becomes to identify strategies that allow us to focus on what should be seen as more pressing issues.

### Our beliefs about self-correcting science need self-correcting

Erroneous belief three is that science self-corrects. Assumptions that published studies are systematically replicated/replicable, or are later identified if they are not, build resistance against reforms. In theory, reproducibility injects quality assurance into the very heart of research. When one adds other traditional safeguards such as professional research norms and peer review, the reliability of research seems well guarded.

However, a growing body of research to check whether scientific results can be reproduced confirms the shortcomings of these safeguards (*Hudson, 2003*; *Allchin, 2015*; *Banobi et al., 2011*; *Zimmer, 2011*; *Twaij et al., 2014*; *Drew, 2019*). We mention just two examples of this research here. The Reproducibility Project: Cancer Biology has been underway for almost five years and originally sought to reproduce 50 critical cancer biology studies (*Couzin-Frankel, 2013*). The project was scaled back to 18 studies, due largely to costs, but also because important details about research methods were unreported in some of the studies the effort sought to reproduce. As

# When errors get corrected, it is more often due to happenstance than any kind of methodical effort

for results, of the first 13 completed replication studies, only five produced results similar to the original studies while the other eight produced either mixed or negative results (*Kaiser, 2018*).

An effort to replicate the findings of 100 experimental studies in psychology journals produced a similarly low rate of replication. Only 36% of the original findings were replicated according to the conventional statistical significance standard of p<0.05 for an effect in the same direction (*Open Science Collaboration, 2015*).

Such findings serve as a vivid wake-up call that alerts us to how easily and how often erroneous research results make their way into print, often in leading journals. Once there, they may linger for years or even decades prior to being discovered (if they are ever discovered) (*Judson, 2004*; *Bar-Ilan and Halevi, 2017*), and may continue to be cited post-discovery (*Steen, 2011*). And when errors get corrected, it is more often due to happenstance than any kind of methodical effort (*Allchin, 2015*). All this is sobering when we consider that erroneous findings can result in potentially dangerous clinical trials (*Steen, 2011*).

Further shaking our confidence in the ability of science to self-correct is how few opportunities there actually are to confirm results. Efforts such as the Reproducibility Project: Cancer Biology notwithstanding, most research sponsors and publishers value, and thus fund and publish, innovative studies rather than research that tries to confirm past findings. And even if sponsors did place higher value on confirmatory studies, the growing complexity of science can make confirmation difficult, or even impossible (*Jasny et al., 2011*). Besides information about study methods and materials possibly not being available, studies may also use novel and/or highly sensitive/volatile study materials (*Hines et al., 2014*), impinge on intellectual property rights (*Williams, 2010*; *Godfrey and German, 2008*), or deal with proprietary data sets (*Peng, 2011*). Thus, even if there was a time in science when there were chances 'to get it

right' or when consensus could emerge, that is no longer the case (*Yarborough, 2014b*).

### Following the rules does not guarantee we are getting it right

Erroneous belief four is that compliance with regulations is capable of solving the problems that gave rise to the regulations themselves. Governments, research sponsors and publishers have gone to great lengths to implement reforms that one hopes contribute to deserved trust. But this is true only to a point; one can follow all the rules, extensive though they may be, and still not get it right (*Yarborough et al., 2009*). We offer efforts to combat research misconduct in the United States as evidence.

The United States Congress, following a series of research scandals, issued a mandate for corrective action to combat falsification, fabrication and plagiarism. This eventually led to a program that endures to this day (*Office of Research Integrity, 2015*), requiring federally funded institutions to investigate allegations of research misconduct. The much larger body of poor-quality science is left completely unaddressed by these government rules. Research shows that about 2% of researchers report engaging in misconduct while fifteen times as many (30%) report having engaged in practices that contribute to irreproducible research (*Fanelli, 2009*); other studies report even higher percentages (*John et al., 2012*; *Agnoli et al., 2017*). Yet, due to the need to follow the rules, resources go overwhelmingly to investigating misconduct. Thus, while such rules bestow quite modest protections to research, they require significant time, energy and money (*Michalek et al., 2010*), and simultaneously provide a false sense of security that problems are being resolved – when in fact they are not (*Yarborough, 2014b*).

## Suggestions to help build cultures and climates that assure deserved trust

If we can find a way to shed these erroneous beliefs, we could become more proactive in showing how we deserve the public's trust. We would not need to start de novo. There are already some proven solutions, as well as promising new recommendations and reforms, that can make inroads on many of the problems identified above. We highlight just a few of them below. Broad implementation of such initiatives could pay valuable dividends. For instance,

**If authors felt safe bringing honest errors to the attention of others, it would encourage much-needed openness about the mistakes that inevitably occur within fields as complex as biomedical research.**

rather than expend extraordinary resources on investigations of misconduct after it has caused damage (*Michalek et al., 2010*), we might instead fund empirical studies of both existing and proposed reforms. In consequence, we could determine which reforms are most capable of strengthening the overall health of biomedical research (*Ioannidis, 2014*).

We recognize that the solutions that we highlight below do not do justice to them as a class, but we do believe they constitute a reasonably representative group. Nor do we mean to suggest that they are without controversy. The main point of our essay, however, is not to provide a thorough review of current and proposed reforms and their individual merits. To do so would focus readers' attention on what changes need to be made in research; our purpose is to explore erroneous beliefs that may prevent sufficient focus on why changes are needed in the first place.

### Publishing reforms: underway but they could be more ambitious

It is encouraging to see that many journals have begun to implement important reform measures. Among the most encouraging is that some now perform rigorous statistical review of appropriate studies, or make such reviews available to peer reviewers or associate editors who request them. Some journals have also modified their instructions to authors in order to improve the reporting of research results. The improved instructions bring transparency to research and aid reproducibility efforts. Recent studies of these modified instructions show that they improve published preclinical study reports, suggesting that even modest journal reforms can work to good effect (*The NPQIP Collaborative group, 2019*; *Minnerup et al., 2016*). It should be noted, though, that the benefits of such

reforms might be small. A recent study showed that a checklist designed to improve compliance with the ARRIVE guidelines had a quite limited effect (*Hair et al., 2018*), showing that having helpful tools is no guarantee that they will be used. Thus, it remains unclear what the ultimate impact of such reform measures might be.

With this evidence in mind, it would be nice if journals were even more ambitious and took on some more novel recommendations. One example is to consider expanding the taxonomy for correcting and retracting publications so that authors can avoid the current stigma around correcting the scientific record (*Fanelli et al., 2018*). This would make it possible to take up a 2016 recommendation to reward authors for self-corrections and retractions (*Fanelli, 2016*). If authors felt safe bringing honest errors to the attention of others, it would encourage much-needed openness about the mistakes that inevitably occur within fields as complex as biomedical research.

### Researcher practices: plentiful recommendations with too few takers

Publisher reforms can only accomplish so much. Most of the improvements that are required to demonstrate how the research community deserves the public's trust need to arise from how research is conducted. A wealth of thoughtful recommendations are already in place, but too many are awaiting widespread adoption. Among the most notable are a set of recommendations for increasing value and reducing waste in biomedical research that appeared as part of a series of articles in *The Lancet* in 2014.

Those recommendations center around several needs: to carefully set research priorities; improve research design, conduct and analysis; improve research regulation and management; reduce incomplete or unusable reports of studies; and make research results more accessible (*Macleod et al., 2014*; *Chalmers et al., 2014*; *Ioannidis et al., 2014*; *Salman et al., 2014*; *Glasziou et al., 2014*; *Chan et al., 2014*). The series has not gone without notice, with more than 46,000 downloads of articles in the series within the first year of publication (*Moher et al., 2016*) and over 900 citations (as of early 2019) in PubMed Central registered articles. Early evidence suggested that the series placed the issues that it addressed on the radar screens of research sponsors, regulators and journals. Disappointingly, academic institutions initially did not seem to pay them much notice (*Moher et al., 2016*). This reinforces our concern

that we need to identify what it is about the mindset of so many in the research community that is currently stifling interest in reform. So long as this lack of interest persists, there is little hope that what we consider the highest impact changes will occur anytime soon. We have two such changes in mind that researchers themselves need to take more of the lead on.

## We need to improve research design and its reporting

Researchers need to pay more attention to research methodology, given its central role in establishing the reliability of published research results. Some journals now encourage this behavior by, for instance, requiring that authors complete checklists to indicate whether or not they have used study design procedures such as blinding, randomization and statistical power analysis. Depending on the journal and type of study, modest to substantial gains in reporting prevalence of study design details are achieved when researchers can complete these requirements (*The NPQIP Collaborative group, 2019*; *Hair et al., 2018*; *Han et al., 2017*). Such improved reporting allows for better assessment of the published literature. Better still would be researchers routinely using universally accepted basic procedures. For example, it is widely acknowledged that for animal studies, randomly allocating animals to groups and blinding experimenters to group allocations is required for sound statistical inference (*Macleod, 2014*).

## We need to increase data sharing

Routine sharing of data should be the new default for researchers, unless there are compelling reasons not to share. Data sharing can, among other things, promote reproducibility, improve the accuracy of results, accelerate research, and promote better risk-benefit analysis in clinical trials (*Institute of Medicine, 2013*). Despite the growing consensus about the value that data sharing brings to research, we must acknowledge that when and how data sharing should occur remains controversial. As recently noted, "[s]ome argue that the researchers who invested time, dollars, and effort in producing data should have exclusive rights to analyze the data and publish their findings. Others point out that data sharing is difficult to enforce in any case, leading to an imbalance in who benefits from the practice – a problem that some researchers say has yet to be satisfactorily resolved" (*Callier, 2019*). Given such issues, it comes as no surprise that compliance with

journal data sharing policies can be lackluster (*Stodden et al., 2018*).

Taking these difficulties into consideration, realistic suggestions to encourage data sharing include: 1) that all journals implement a clear data sharing policy (*Nosek et al., 2015*) that allows reasonable flexibility to take into account cases when data cannot be shared because of ethical or identity protection concerns, or that allow 'embargo' periods during which data are not shared (*Banks et al., 2019*); 2) that journals systematically require data sharing during the review process, to help reviewers to evaluate the results (this would have the additional benefit of meaning that no additional effort is required afterward to make the data public); 3) that training courses in Responsible Conduct of Research (RCR) include methods to de-identify study participants and aggregate their results (a major prerequisite to data sharing [*Banks et al., 2019*]); and 4) the creation of awards for researchers who promote data sharing (*Callier, 2019*).

Finally, we need to know whether improved methodology and increased data sharing are really leading to reproducible research. Unfortunately, we could not locate studies that have addressed this question, making this an important line of future research.

### Institution level practices: promising and proven remedies looking for suitors

When it comes to institutional practices that could strengthen the trustworthiness of research, surely the holy grail would be to better align researcher incentives with good science (*Ware and Munafò, 2015*). This would be a heavy lift since it would involve changes to how institutions collectively approach recruitment, tenure and promotion. Rather than relying upon current surrogates such as bibliometrics for assessing faculty productivity and success (*McKiernan, 2019*), they would need to use more direct measures of good science. A workshop involving research quality and other experts was convened in Washington DC in 2017 to explore what such measures might be and how they might be used. It identified six key principles that institutions could embrace to effect such a transition (*Moher et al., 2018*), but their effectiveness remains untested as they have yet to be implemented. It is worth noting, however, that at least one institution – the University Medical Center Utrecht – has tried to reengineer how it assesses its research programs and faculty in order to better align incentives

with good science. In the words of the champions of that change initiative, they are learning how to better "shape the structures that shape science…[to] make sure that [those structures] do not warp it" (*Benedictus et al., 2016*).

There are smaller scale reforms that institutions could also embrace to help ensure high quality standards in research. For example, there are many innovative practices that institutions could currently use to prevent problems, but are not. Perhaps the most obvious one is a research data audit. Akin to a finance audit, a research data audit is meant to check that published data are "quantifiable and verifiable" by examining "the degree of correspondence of the published data with the original source data" (*Shamoo, 2013*). First proposed at scientific conferences in the 1970 s, (*Shamoo, 2013*) and later in print in *Nature* in 1987 (*Dawson, 1987*), such audits "would typically require the examination of data in laboratory notebooks and other work sheets, upon which research publications are based" (*Glick, 1989*). Advocates argue that data audits should be routine in as many settings as possible. This would provide a double benefit; it would help to deter fraud on the one hand and promote quality assurance on the other (*Shamoo, 2013*).

The FDA and the United States Office of Research Integrity currently conduct such audits 'for cause' when misconduct or other misbehaviors are suspected. The FDA also uses them for certain new drugs deemed to be potentially 'high risk.' Although most current audits typically review the proper use of specified research procedures, there is no reason that they could not also be used to encourage the proper generation and use of actual data (*Shamoo, 2013*).

Critical incident reporting (CRI) is another promising prevention practice. It can be used to uncover problems, that, if left unchecked, might prove detrimental to a group's research or reports about their research. Open software exists for implementing such a system. Accessed anonymously online, the system prompts users to report in their own terms what happened that is of concern to them. Experts can then promptly analyze incidents to see what systems changes might prevent future recurrences. The first adopters of such a system report that it "has led to the emergence of a mature error culture, and has made the laboratory a safer and more communicative environment" (*Dirnagl et al., 2016*).

The same opportunity pertains to two other successful problem reduction methods: root cause analysis (RCA) and failure modes and effects analysis (FMEA) (*Yarborough, 2014a*). RCA examines past near misses and problems in order to identify their main contributors. FMEA anticipates ways that future concerns might occur and prioritizes the severity of negative consequences if they do occur (for example, in aviation one might compare increased fuel consumption by a plane versus the catastrophic failure of a wing). The most critically needed preventive measures can then be targeted to avoid severe problems occurring in the first place.

RCA and FMEA have both been used to good effect across a wide spectrum of industries and endeavors, including the pharmaceutical industry and clinical medicine. Their track record clearly shows that they can be used to reduce medication, surgical and anesthesia errors, and ensure quality in the drug manufacturing process. Both these methods lend themselves most easily to manufacturing and engineering settings, but their successes suggest they also warrant testing for use in research. In particular, they may improve the human factors that can lead to avoidable problems, especially in team-based science settings where geographic dispersion and distributed expertise are the norm (*Yarborough, 2014a*; *Dirnagl et al., 2016*).

It seems clear that data audits, CRI, RCA, and FMEA each have tremendous potential for improving research: potential that, like the above publishing reforms and researcher practices, has gone largely untapped to this point. We worry that the four erroneous beliefs that we have highlighted are blunting curiosity about the health of biomedical research, and are thereby preventing the adoption of a more proactive stance toward quality concerns. Hence, a critical next challenge is learning how to erode the appeal of these beliefs.

One strategy that we think is particularly worth considering is education. A wider appreciation of evidence that demonstrates the range and extent of quality concerns in research, combined with evidence about how few of them stem from research misconduct, should diminish belief that a few bad apples are our biggest problems. A placeholder for this education is already in place. RCR education is now firmly ensconced in many graduate and postgraduate life sciences courses and could naturally incorporate modules that tackle the erroneous beliefs head on.

We should note, however, that this strategy is far from perfect, given longstanding concerns about the effectiveness of RCR curricula

**There are plenty of thoughtfully tailored recommendations that have not yet resulted in the improvements to research they are surely capable of producing**

(*Antes et al., 2010*; *Presidential Commission for the Study of Bioethical Issues, 2011*) and the fact that sponsors who mandate RCR instruction, like the National Institutes of Health (NIH) and the National Science Foundation (NSF) in the United States, often stipulate content that needs to be covered by it. The latter challenge need not be insuperable, though, since both NIH and NSF also encourage innovation and customization of RCR learning activities. Using RCR education as a vehicle for fostering improved quality in research may also help to make such instruction appear more relevant to the careers of learners.

As an example, RCR sessions could examine the scientific record on self-correction. The aforementioned cancer and psychology replication projects would surely warrant consideration, but we think that an equally relevant and highly illustrative case study showing how this might be done is a recently published study (*Border et al., 2019*) about the lasting detrimental impact of a 1996 study about the SLC6A$_4$ gene on depression research (*Lesch et al., 1996*). This publication spurred at least an additional 450 published ones, consumed millions of dollars, and controversy about it continues to this day (*Yong, 2019*). Such case studies can drive home multiple lessons because they simultaneously show how science cannot be relied upon to self-correct in a timely or efficient way and that regulations often fail to touch upon matters critical to the health of research.

## Conclusion

Readers may be tempted to dismiss the foregoing analysis of erroneous beliefs as mere personal observations. They may prefer instead either hard data about how research measures up against metrics that contribute to deserving trust. Or they may wish for yet another round of study design and data analysis

recommendations capable of solving the broad range of ills currently diminishing the quality of research. The recommendations would plot the path to progress while the data would make our pace of progress apparent to all.

As we have tried to make clear, there are plenty of thoughtfully tailored recommendations that have not yet resulted in the improvements to research they are surely capable of producing – simply because there has been too little uptake of them. Nor, for that matter, is there any shortage of calls to arms and manifestos, including those from some of the most eminent scholars and leaders in biomedical research (*Alberts et al., 2014*; *Munafò et al., 2017*). Since these have had such little effect so far, especially at the institutional level, it is not clear why we would expect yet more recommendations to enjoy a better reception. Besides, many questionable research practices are hidden from view. For example, inconvenient data points, or even entire experiments, are at times ignored (*Martinson et al., 2005*); data are added to experiments until desired p-values are obtained (*Simmons et al., 2011*); and unreliable methods are used when randomizing animals in studies (*Institute for Laboratory Animal Research Roundtable on Science and Welfare in Laboratory Animal Use, 2015*). Because these behaviors are hidden, traditional metrics are unlikely to capture their extent or their influence on the trustworthiness of research.

These behaviors notwithstanding, 'open science' practices would be one way to increase confidence in research results that could also provide metrics of trustworthiness. For example, some questionable research practices, such as p-hacking (*Head et al., 2015*), could be detected more easily by requiring that data and analysis code be publicly available in all but the most exceptional circumstances. Indeed, one group has called for traditional institutional performance metrics such as impact factor and number of publications to be replaced with open science metrics (*Barnett and Moher, 2019*). Although measurable open science would not eliminate questionable research practices, it would move biomedical research toward increased accountability.

Open science practices are still no panacea, however, for all the quality concerns we have highlighted here. What is most needed at this juncture is a collective focus on deserving trust. Such a focus could make researchers and the leaders of research institutions more receptive to reform efforts. The four erroneous beliefs we

have discussed surely hinder that collective focus, and thus deter the research community from adopting reforms that can secure the public's trust – which is vital to biomedical research.

Mark Yarborough is in the Bioethics Program, University of California, Davis, Sacramento, CA, United States

mayarborough@ucdavis.edu

https://orcid.org/0000-0001-8188-4968

Robert Nadon is in the Department of Human Genetics, McGill University, Montreal, Canada

David G Karlin is an independent researcher based in Marseille, France

Author contributions: Mark Yarborough, Robert Nadon, David G Karlin, Conceptualization, Writing— original draft, Writing—review and editing

Competing interests: The authors declare that no competing interests exist.

Funding

The authors declare that there was no funding for this work

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
