## [Decision Letter]

Thank you for submitting your article "Four erroneous beliefs thwarting more trustworthy research" for consideration by *eLife*. Your article has been reviewed by three peer reviewers, and the evaluation has been overseen by Emma Pewsey (Associate Features Editor) and Peter Rodgers (Features Editor). The following individuals involved in review of your submission have agreed to reveal their identity: Malcolm R MacLeod (Reviewer #1); Martin Michel (Reviewer #2); Olavo B Amaral (Reviewer #3).

The reviewers have discussed the reviews with one another and I have drafted this decision to help you prepare a revised submission.

Please note that the reviewers have raised a number of discussion points. You do not need to address these in the manuscript itself, but you may wish to respond to them in your author response, which will appear on the *eLife* website along with your article and the decision letter.

Summary:

The article nicely summarises most of the major problems surrounding trust in biomedical research. While it is not the first article on this topic, the authors have written it up in a very refreshing and enjoyable way.

Essential revisions:

1) Throughout the text it appears to be more accurate to refer to "irreproducible" instead of "erroneous" research results. A result that is not reproduced might be a fully accurate description of a real experiment, but one that either describes a chance finding, a finding that is dependent on very specific experimental conditions, or it could indeed be erroneous (i.e. fabricated, falsified, or due to an erroneous analysis or method). If you feel it is important to use the term "error", please clearly define what it means.

One suggestion from the reviewers to help you reframe the discussion to avoid a focus on "errors" is to think of research practices on a continuum, from the unforgivable to the perfect. At each point, the value of research can be reduced by making "errors" (be these deliberate or inadvertent, or even just consequences of the limitations of the methodologies used) or enhanced by adopting better practice (e.g. open data). For instance, the value of a non-randomised animal study is enhanced if this is clearly stated (because you can interpret the findings accordingly), and the value of any evidentiary claim is enhanced if it was asserted as the primary outcome measure in an a priori study protocol. Strategies that incorporate these different approaches might produce marginal improvements across the continuum.

2) Although the recommendations that follow the discussion of the "four erroneous beliefs" are all sensible, most of them do not address the beliefs directly. Please discuss in greater depth how the beliefs thwart attempts at reform, the approaches that could be taken to change them, and any further barriers that prevent this from happening. Ideally this discussion will include some empirical evidence and concrete examples.

Other discussion points:

1) In reviewer #2's experience of teaching on better reproducibility by more vigorous study design, data analysis and reporting, the PhD students in the class immediately see the point. However, they often come back and ask for advice when their supervisor dismisses concerns about reproducibility with "we've always done it that way" and "everybody does it that way". Have you also found that young researchers are more open to change than established ones?

2) When discussing beliefs 1 and 2 ("it's about the science, not the scientists" and "we need to focus on the health of the orchard, not just the bad apples"), it could be worth discussing the fact that much of the training in research ethics over the last decades has revolved around misconduct. This is likely to reinforce these two beliefs – and, according to the authors' view, might preclude scientists from seeing the problem as their own (as misconduct and ethics breaches are rarely something that one will admit to). Perhaps framing research ethics training around the science (e.g. irreproducible research and its consequences), rather than the scientists, might help in overcoming these beliefs.

3) The authors mention at various points that there is "reluctance within the research community to implement the reforms", that "there has been too little uptake of them" and that "calls to arms have had little effect so far". The authors could discuss this issue in more depth and provide evidence to show that this is the case. What constitutes "little effect"? Some change does appear to be happening, albeit slowly. Although data and code sharing are still subpar, they are likely to be on the rise (e.g. Campbell et al., 2019 at doi: 10.1016/j.tree.2018.11.010), as are open access (Piwowar et al., 2018 at doi: 10.7717/peerj.4375) and preprint deposition (Kaiser, 2017 at doi: 10.1126/science.357.6358.1344).

4) The authors speak of "reluctance within the research community to implement the reforms". But is the community really reluctant, or are many people unaware of the problem or unfamiliar with the possible solutions and necessary reforms? Similarly, the authors state that "these beliefs are blunting curiosity about the health of biomedical research". The rising interest in the subject over the last few years, suggests that inertia in taking necessary changes and a feeling of non-responsibility and/or powerlessness towards them ("it's the system, not me") might be more important than lack of acknowledgement. How do these different factors contribute and interact to prevent necessary changes?

5) On Belief 4, the authors argue that the fact that investment in tackling misconduct has failed to prevent irreproducible research is evidence that "following the rules doesn't guarantee we are getting it right". But couldn't it be the case that this effort has focused on the major rules (e.g. do not falsify, fabricate or plagiarize) but has ignored the (many more) minor ones that are just as vital for published research findings to be reproducible (e.g. use adequate power, differentiate exploratory and confirmatory research, avoid p-hacking and HARKing)? The failure to prevent bad science could be because there are too few rules, rather than because following the rules doesn't work.

6) On the issue of science not being systematically self-correcting, it might be worth mentioning the high prevalence of failed replication attempts that are not published, most commonly due to the authors of replications not attempting to publish them – see Baker et al., 2016 (doi: 10.1038/533452a) for a survey-based indication of that.

7) "… rather than research that tries to confirm past findings." One possibility to increase confirmation is to place higher value on confirmatory studies of past findings, but the other would be to raise the threshold for publication in the first place in some instances (e.g. requiring preregistration or independent confirmation, see Mogil and Macleod, 2017 – doi: 10.1038/542409a). As there are arguments for both sides, it could be worth touching on this point.

8) Subsection “Publishing Reforms: underway but they could be more ambitious”: You could discuss whether, with such a large number of journals (in which peer review varies widely in quality), and preprints making important headway in biology, we should expect the main source of quality control to come from journals. Note that arguments have been made in the opposite direction (e.g. removing barriers to publication to diminish its reward value and make 'publish or perish' senseless – e.g. Nosek and Bar-Anan, 2012 http://dx.doi.org/10.1080/1047840X.2012.692215 and others).

9) It sounds somewhat incongruent to state that there has been little change in research practices, while at the same time arguing that many of these practices cannot be measured. A counterargument to the statements that metrics are unable to counter irreproducibility is that to change incentives in order to foster trust, assessing what kind of research is more "deserving" of trust is important – thus, good metrics are perhaps precisely what is needed to build up trust. Sharing of data and analysis code, for example, can help to assess whether p values have been hacked (as they allow for reanalysis of the data using other methods). Thus, using appropriate sharing of data as a metric is likely to improve some of these issues.

---

## [Author Response]

Essential revisions:1) Throughout the text it appears to be more accurate to refer to "irreproducible" instead of "erroneous" research results. A result that is not reproduced might be a fully accurate description of a real experiment, but one that either describes a chance finding, a finding that is dependent on very specific experimental conditions, or it could indeed be erroneous (i.e. fabricated, falsified, or due to an erroneous analysis or method). If you feel it is important to use the term "error", please clearly define what it means.One suggestion from the reviewers to help you reframe the discussion to avoid a focus on "errors" is to think of research practices on a continuum, from the unforgivable to the perfect. At each point, the value of research can be reduced by making "errors" (be these deliberate or inadvertent, or even just consequences of the limitations of the methodologies used) or enhanced by adopting better practice (e.g. open data). For instance, the value of a non-randomised animal study is enhanced if this is clearly stated (because you can interpret the findings accordingly), and the value of any evidentiary claim is enhanced if it was asserted as the primary outcome measure in an a priori study protocol. Strategies that incorporate these different approaches might produce marginal improvements across the continuum.

To address these comments, we have dropped most of the uses of the term errors and mistakes and, unless the context for their use is clear, use instead such terms as “problems” or “unreliable research.” We thank the reviewer for helping us to see how vague the terms “errors” and “mistakes” can be. We gave a lot of thought to alternative phrasing, including using “irreproducible” as suggested but in the end decided against using that particular term. We explain our thinking here. First, our original reason for using terms like error and mistake was because we wanted to be clear to readers that our main focus was not research misconduct. As we explain later in the text, research misconduct is already consuming enough of the oxygen in the room that could be used instead to combat more prevalent problems. Second, while “irreproducible research” certainly covers much, perhaps even most, of the terrain that is in fact our focus, we worried that phrase may be off-putting for some readers and a bit narrow for our purposes. Despite the widely-cited surveys documenting that most researchers believe there is a “reproducibility crisis,” many in the research community nevertheless reject that narrative and these are the very readers we are most hoping to engage in our essay. Third, we thought that, at least technically speaking, framing everything in terms of irreproducible research might perhaps also be a bit vague, given the distinct ways that results might fail to reproduce. (Here we have the Goodman/Fanelli/Ioannidis STM discussion in mind.) We think that terms such as “unreliable research”, “problems”, and “erroneous research results” avoid these issues while accurately conveying our intent so those are the new terms we settled on. We made similar changes near the end of the manuscript. Finally, we found the suggestion to reference a continuum of research practices especially helpful, so we incorporated it into the manuscript near the outset.

2) Although the recommendations that follow the discussion of the "four erroneous beliefs" are all sensible, most of them do not address the beliefs directly. Please discuss in greater depth how the beliefs thwart attempts at reform, the approaches that could be taken to change them, and any further barriers that prevent this from happening. Ideally this discussion will include some empirical evidence and concrete examples.

This is a somewhat complex request to address. The recommendations following our discussion of the erroneous beliefs were added at the request of the editors, which they wanted to see prior to deciding to send the manuscript out for review. So, we feel like they should remain in the manuscript. The reviewers are correct in pointing out, however, that they do not directly mitigate the erroneous beliefs themselves. Thus, we have added a discussion of educational activities to show how one could take on the erroneous beliefs more directly, as well as an example as to how this might be done. We are not sure how to integrate into the manuscript, though, more in-depth discussion about exactly how the four beliefs are thwarting attempts at reform. The impetus for the manuscript was curiosity about why so many reforms that target a range of concerns have yet to be acted upon by too many in the research community, a question that we acknowledge mainly affords speculation at this point. We attribute this inattention to a failure to fully appreciate the need for these reforms, despite the wealth of published evidence establishing how much they are needed. What might account for this inattention is surely a complex range of factors that it would be hard to study and definitively quantify, but it seems reasonable to assume that a certain mindset, characterized in part by the erroneous beliefs, is part of that range. So, we wanted to try to tease out some of the components about that mindset in order to spur discussion about them. We continue to believe that an ensuing discussion about them should prove valuable even in the absence of targeted empirical study about how widespread they are and how they may be specifically thwarting individual recommendations. We think that highlighting the general issues covered by the four beliefs can better spur reflection and conversation than would an effort to try to tie specific beliefs to the uptake rate of specific reforms in the absence of studies designed to test those ties. We trust that this more general approach is acceptable for articles in the “Features” section of the journal.

Other discussion points:

We have given a lot of thought about how/whether to address in the manuscript the discussion points below and decided to only make one minor addition and would like to briefly explain our thinking here. While all of the suggestions have a lot of merit, we worried that discussing them in the manuscript would focus readers’ attention on *what* changes need to be made in research whereas the broad focus of our manuscript is *why* changes need to be made. We have inserted new text in the manuscript to this effect. We would very much like to keep that the focus and we worry that too much discussion about actual proposed changes will prompt readers to dwell on whether they agree with those particular changes rather than the need for change itself. Reforms preceded by widespread recognition of the need to change will arguably fare better than ones undertaken when there is no consensus about the need for them in the first place. That is why we prefer not to include more discussion than we already have about representative reforms to tackle various problems. We know that the reviewers are very familiar with the extensive landscape of current reforms underway and the many candidates for additional ones. We also hope that the reviewers and editors will agree that global considerations about the health of the research enterprise and the current systems that shape it have value. We wanted to preserve this more global perspective and were a bit worried that too much discussion of particular reforms will draw attention away from the broader considerations. Our edits are an attempt to strike this balance. We trust that this will be acceptable to the reviewers and editors.

1) In reviewer #2's experience of teaching on better reproducibility by more vigorous study design, data analysis and reporting, the PhD students in the class immediately see the point. However, they often come back and ask for advice when their supervisor dismisses concerns about reproducibility with "we've always done it that way" and "everybody does it that way". Have you also found that young researchers are more open to change than established ones?2) When discussing beliefs 1 and 2 ("it's about the science, not the scientists" and "we need to focus on the health of the orchard, not just the bad apples"), it could be worth discussing the fact that much of the training in research ethics over the last decades has revolved around misconduct. This is likely to reinforce these two beliefs – and, according to the authors' view, might preclude scientists from seeing the problem as their own (as misconduct and ethics breaches are rarely something that one will admit to). Perhaps framing research ethics training around the science (e.g. irreproducible research and its consequences), rather than the scientists, might help in overcoming these beliefs.

The discussion we added includes some of these same points. Also, with respect to Other discussion point 1 above, in the experiences of the lead author, he finds much more student interest in his research ethics class in research quality and reproducibility issues than was the case as recently as 3-4 years ago. However, given the length of our manuscript and the anecdotal nature of the observation, we chose not to include this in the body of the paper.

3) The authors mention at various points that there is "reluctance within the research community to implement the reforms", that "there has been too little uptake of them" and that "calls to arms have had little effect so far". The authors could discuss this issue in more depth and provide evidence to show that this is the case. What constitutes "little effect"? Some change does appear to be happening, albeit slowly. Although data and code sharing are still subpar, they are likely to be on the rise (e.g. Campbell et al., 2019 at doi:10.1016/j.tree.2018.11.010), as are open access (Piwowar et al., 2018 at doi:10.7717/peerj.4375) and preprint deposition (Kaiser, 2017 at 10.1126/science.357.6358.1344).

We have also chosen not to add any extensive additional text, though we did add “especially at the institutional level” to moderate our claim somewhat. We trust that the reviewers and editors will be ok with this decision. One complicating factor here is that much of the evidence would be found in the blogosphere and popular press, where comments make it clear that there is still a lot of resistance in many quarters about both the extent and severity of problems. See, for example, https://www.theatlantic.com/science/archive/2018/11/psychologys-replication-crisis-real/576223/?utm_source=feed and the skeptics addressed there. We can also point to numerous conversations of our own and of colleagues who regularly get pushback when the topic is problems in research. Also, we think that we have made reference in the manuscript to several examples of reforms in place and the extent of their impact to date.

4) The authors speak of "reluctance within the research community to implement the reforms". But is the community really reluctant, or are many people unaware of the problem or unfamiliar with the possible solutions and necessary reforms? Similarly, the authors state that "these beliefs are blunting curiosity about the health of biomedical research". The rising interest in the subject over the last few years, suggests that inertia in taking necessary changes and a feeling of non-responsibility and/or powerlessness towards them ("it's the system, not me") might be more important than lack of acknowledgement. How do these different factors contribute and interact to prevent necessary changes?

Again, we have chosen not to address this suggestion specifically. The reviewer is no doubt correct to point out that some important changes are afoot. It is also worth noting that reluctance may not be the preferred term to use here since, as the reviewer suggests, it might be a feeling of powerlessness, lack of awareness, or something else. But we remain comfortable with the term “reluctance.” Our collective sense is that the impetus for reform is still confined to a significant degree within the meta-research community, despite the years, at times even decades, e.g., continued use of mislabeled cell lines and citations of studies known to involve them, of publicizing and discussing the problems. This gap is what we are curious to try to understand. Hence this essay speculating that the 4 erroneous beliefs we identify are likely contributors to it. Perhaps one finds it easy to deflect concern about irreproducible or otherwise unreliable research if one is confident that science self-corrects. If it does, then there is less need to worry about methodologic quality or reproducibility. Or, why worry about bias if there are now required disclosures about financial interest? We think such erroneous beliefs are at least as contributory to the current disappointing pace of reform as is lack of awareness or feelings of powerlessness and as such warrant the consideration of readers. And please note that our discussion about RCR does pertain to the awareness issue.

5) On Belief 4, the authors argue that the fact that investment in tackling misconduct has failed to prevent irreproducible research is evidence that "following the rules doesn't guarantee we are getting it right". But couldn't it be the case that this effort has focused on the major rules (e.g. do not falsify, fabricate or plagiarize) but has ignored the (many more) minor ones that are just as vital for published research findings to be reproducible (e.g. use adequate power, differentiate exploratory and confirmatory research, avoid p-hacking and HARKing)? The failure to prevent bad science could be because there are too few rules, rather than because following the rules doesn't work.

Again, we have chosen not to address this suggestion specifically. While we share much of this reviewer’s diagnosis, we also believe that researchers have to internalize a deep commitment to proper research methodology and we are skeptical that assuring the use of proper methods is best accomplished by having more rules for researchers to follow. We think that a more productive approach is having more virtuous researchers in the Aristotelian sense (habitually doing things in the right way with the proper motivations and the right reasons) and we are suggesting that unhelpful beliefs like the ones we have highlighted are a possible major culprit hindering a stronger allegiance to proper research methodology that deserves greater focus.

6) On the issue of science not being systematically self-correcting, it might be worth mentioning the high prevalence of failed replication attempts that are not published, most commonly due to the authors of replications not attempting to publish them – see Baker et al., 2016 (doi: 10.1038/533452a) for a survey-based indication of that.

We think that devoting space within the article to this suggestion would take us a bit off target in that there are so many causes that hinder self-correction, including failure to publish results, which is already discussed and documented in the literature, some of which we have cited, while our focus is the broad issue of science not self-correcting.

7) "… rather than research that tries to confirm past findings." One possibility to increase confirmation is to place higher value on confirmatory studies of past findings, but the other would be to raise the threshold for publication in the first place in some instances (e.g. requiring preregistration or independent confirmation, see Mogil and Macleod, 2017; doi:10.1038/542409a). As there are arguments for both sides, it could be worth touching on this point.

We completely agree that this would be another possibility. Our response here is similar to the immediate one above. We are a bit reluctant to extend our discussion because we think the current discussion already supports the major points we wanted to make. In addition, changing the nature of journals and what they publish, as the reviewer suggests, is a long and heterogenous process and shaking faith that science self-corrects, which is part of our aim here, might prove to be a useful accelerant to that process. Thus, rather than focus on what changes are most needed, we have chosen instead to try to spur greater reflection that could help show why changes are needed in the first place.

8) Subsection “Publishing Reforms: underway but they could be more ambitious”: You could discuss whether, with such a large number of journals (in which peer review varies widely in quality), and preprints making important headway in biology, we should expect the main source of quality control to come from journals. Note that arguments have been made in the opposite direction (e.g. removing barriers to publication to diminish its reward value and make 'publish or perish' senseless – e.g. Nosek and Bar-Anan, 2012 http://dx.doi.org/10.1080/1047840X.2012.692215 and others).

This suggestion has a lot to recommend it as well but we only have so much space and we had to have some parameters for our discussion of publisher reforms so we chose to limit those to the kinds of incremental reforms we highlighted, as opposed to a radical rethink of scientific publishing. We trust this demarcation will be acceptable to the reviewers and editors.

9) It sounds somewhat incongruent to state that there has been little change in research practices, while at the same time arguing that many of these practices cannot be measured. A counterargument to the statements that metrics are unable to counter irreproducibility is that to change incentives in order to foster trust, assessing what kind of research is more "deserving" of trust is important – thus, good metrics are perhaps precisely what is needed to build up trust. Sharing of data and analysis code, for example, can help to assess whether p values have been hacked (as they allow for reanalysis of the data using other methods). Thus, using appropriate sharing of data as a metric is likely to improve some of these issues.

We like this idea and have modified the text accordingly.